# Photothermal-Triggered Shape Memory Polymer Prepared by Cross-Linking Porphyrin-Loaded Micellar Particles

**DOI:** 10.3390/ma12030496

**Published:** 2019-02-06

**Authors:** Wangqiu Qian, Yufang Song, Dongjian Shi, Weifu Dong, Xiaorong Wang, Hongji Zhang

**Affiliations:** 1Key Laboratory of Synthetic and Biological Colloids, Ministry of Education, School of Chemical and Material Engineering, Jiangnan University, Wuxi 214122, China; 6160608003@vip.jiangnan.edu.cn (W.Q.); 6170608022@stu.jiangnan.edu.cn (Y.S.); djshi@jiangnan.edu.cn (D.S.); 1984111521@126.com (W.D.); 2College of Chemistry, Chemical Engineering and Environmental Engineering, Liaoning Shihua University, Fushun 113001, China

**Keywords:** porphyrin, micellar particle, photothermal, shape memory polymer

## Abstract

In this work, we fabricated porphyrin-loaded shape memory polymer (SMP) film by cross-linking micellar particles prepared by co-assembly of porphyrin compounds and amphiphilic macromolecules formulated by copolymerization of 2-butoxy ethanol (BCS), methyl methacrylate (MMA), butyl acrylate (BA) acrylic acid (AA), and diacetone acrylamide (DAAM). The experimental results revealed that this film was able to respond to the red light in terms of photothermal effect enabled by the porphyrin filler. The photothermal-triggered shape memory behaviors of the film were further examined in detail. It was noteworthy that this material was expected to have potential applications in the biomedical field due to the excellent biocompatibility of the porphyrin filler and the red-light source, which was optimal and safe enough for biomedical treatment.

## 1. Introduction

Shape memory materials are stimuli-responsive materials that can be processed into a temporary shape, fixed and retained to transform from a temporary shape back to a permanent shape through external stimuli, such as physical or chemical stimuli. Such a transformation is termed the shape memory effect. The external stimuli specifically include heat [1], light [2,3], electricity [4,5,6], magnetic field [7,8], solvent [9,10], pH [11], moisture [12,13,14], and the like. The main advantages of using light to induce shape transitions are that (1) non-contact remote wireless activation is easily achievable; (2) spatial control with masking or holographic exposure is possible; and (3) recovery depends on the light of selective wavelengths, and shape memory can only be triggered when SMP is illuminated with light of an appropriate wavelength. In recent years, light-activated SMP has been increasingly attracting attention from researchers by virtue of its unique features. The use of light as a stimulus for shape memory effect will further expand the applications of SMPs, especially in the field of biomedicine.

Light-active SMPs need be equipped with photoactive functional groups or light-absorbed fillers. Light-active SMPs can be divided into two types, by photochemical reaction and photothermal effect, according to different memory mechanisms. In photochemical reactive SMPs, the light has been used in the systems incorporating light-sensitive moieties (e.g., cinnamic acid [3,15,16], azobenzene [17,18], etc.) to induce chemical changes (crosslinking/de-crosslinking, photo-isomerization) for shape memory and recovering. However, first of all, the molecular rearrangements that occur are usually not 100% reversible or the chemical reactions are not suitable for all polymer systems. Secondly, the excitation light of most chemical changes is high-energy ultraviolet light, which is not conducive to the application of SMPs as biological implants. In contrast, SMPs based on photothermal effect are more "universal". Simply, the SMPs matrix incorporates some photothermal conversion fillers (gold nanoparticles (AuNPs) [19], gold nanorods (AuNRs) [20], carbon nanotubes (CNT) [11,21], and graphene [22], etc.) that act as strong light absorbers and further convert light energy into heat efficiently through non-radiative energy decay while having high efficiency of transferring it to the polymer matrix. Indirectly, regional shape memory can be induced by local Joule heating. In particular, some practical applications of nano-filler have been reported, such as gold nanoparticles, which have been used in tumors’ thermal ablation. [23,24,25]. Unfortunately, these fillers mentioned above retain more or less toxic residues in the body after being metabolized, thus leading to concerns on their safety and long-term fate in vivo, and limiting the applications of photothermal-based SMPs in the biomedical field. In the present article, we attempt to choose an appropriate light-absorbing filler inside SMPs, with high photothermal conversion efficiency and good biocompatibility, as a photo-active nano-heater.

In the course of long-term research, scientists have discovered a light-harvesting molecule-porphyrin, which is widely found in natural animal bodies (iron-porphyrin, copper-porphyrin) and plants (magnesium-porphyrin, cobalt-porphyrin). The light absorption capacity of porphyrin-loaded nanovesicles (porphysomes) is similar to that of gold nanoparticles, and the absorbed energy is able to be released as heat due to fluorescence quenching, providing exceptional photothermal performance as a nano-heater. Surprisingly, works on photothermal therapy (PTT) reagents of this group show particular advantages of the liposomal nanostructure of porphyrin lipids, including outstanding biodegradability and biocompatibility [26]. All of these results indirectly illustrate the desired high photothermal efficiency and good biocompatibility of porphyrin.

In this study, we use the eco-porphyrin as a light-to-heat convertor to construct a photoactive "adipoid" nanostructure, that is, porphyrin-loaded micellar nanoparticles by co-assembly of the porphyrin compounds and amphiphilic macromolecules in selective solvent (water). On this basis, the red-light-responsive porphyrin-loaded SMP film was prepared by cross-linking these nanoparticles. The heat generated by the red-light irradiation can release the frozen molecular chains and recover SMP from a temporary to permanent shape.

## 2. Materials and Methods

### 2.1. Materials

All solvents and reagents were purchased from Shanghai Macklin Biochemical Co., Ltd., (Shanghai, China) and used in experiments without further treatment

### 2.2. Synthesis

The photothermal-sensitive shape memory polymer film loaded with porphyrin was prepared by cross-linking porphyrin-loaded micellar particles containing reactive groups with adipic dihydrazide (ADH). Specifically, a mixture obtained by thoroughly mixing the micellar emulsion with ADH was cast on a glass slide and slowly dried to remove water for one week at room temperature. Details of synthesis involved in this experiment are shown in Scheme 1.

Porphyrin-loaded micelles containing reactive groups were fabricated by a process of assembling in solution (see Table 1 for proportion). In brief, porphyrin and carbonyl-sided copolymer (prepared as described following) were uniformly dispersed in a polar solvent (such as tetrahydrofuran), preheated to 55 °C and the PH adjusted to 7–8. The solution was constantly mixed while required amounts of H_2_O were slowly added. Porphyrin-loaded micelles were yielded through removing of THF (Tetrahydrofuran) solvent by slow evaporation.

As an example of the carbonyl-sided copolymer, the synthesis of MBDA copolymer was described: A mixture of 5 mL 2-butoxy ethanol (BCS), 22 mL (20.77 g) methyl methacrylate (MMA), 11 mL (9.29 g) butyl acrylate (BA), 1 mL (1.05 g) acrylic acid (AA), 2 g diacetone acrylamide (DAAM), and 0.81 mL (0.84 g) tert-butyl peroxybenzoate (TBPB), which acts as an initiator, were slowly charged into a 250 mL three-necked flask fitted with a constant pressure funnel, and applying stirring and condensation. The solution was stirred for 4.5 h at the constant temperature (145 °C) to complete the reaction. The received product was precipitated in diethyl ether to remove the residual monomers to obtain the desired MBDA copolymer.

### 2.3. Characterizations

In this paper, the SMP films loaded with porphyrin at four different concentrations were prepared. The size of the sample was 75 × 25 mm and the thickness was 0.3 mm. The polymerizing effect of synthesized MBDA copolymer was detected with gel permeation chromatography (GPC, Waters 1515, USA) and the structure units were confirmed by nuclear magnetic resonance hydrogen spectrum (^1^H-NMR, Bruker AVANCE III HD 400 MHz, Switzerland) and Fourier transform-infrared (FT-IR) spectroscopy (Thermo-Nexus 470, USA). Dynamic light scattering (DLS, ZS90, UK) was carried out to investigate the sizes of porphyrin-loaded micelles and their distribution. Absorption ability for light was investigated by UV–Vis absorption spectrum (Agilent Cary 5000, USA) in the wavelength range 350–850 cm^−1^. Glass transition temperatures (*T*_g_) of pure SMP and porphyrin-loaded SMP were determined to set actuation temperature by using differential scanning calorimetry (DSC, TA-Q2000, USA). Finally, under common conditions, the shape memory behavior of light-active porphyrin-loaded SMP was investigated. In this experiment, the film heated to 65 °C (*T* > *T*_g_) was folded and then cooled to room temperature, with external force held to fix the temporary shape. When the bent position was exposed to laser, the film began to unfold due to the photothermal effect caused by porphyrin. During this process, the effect of different porphyrin doses on photothermal effect of SMP was indirectly explored by investigating the relationship between the unfolded angle of the film and the cumulative exposure time. As a comparative test, a film without TPP (Tetraphenylporphyrin) was made into a temporary shape in the same manner, and laser irradiation was received under the same conditions.

## 3. Results and Discussion

### 3.1. Characterizations of MBDA copolymer

Figure 1 presents the successful synthesis of MBDA copolymer. The molecular weight and polydispersity (*M*_w_/*M*_n_) of the copolymer was measured by means of gel permeation chromatography (GPC), with reference polystyrene as the calibration standard and THF as mobile phase (flow rate: 1.0 mL/min). Figure 1a shows the GPC curve of different polymerization time of 2, 3, 4 h, and the molecular weight gradually increased with the extension of reaction time. At 3 h, the average molecular weights were 62000 g/mol (*M*_w_) and 23000 g/mol (*M*_n_). Otherwise stated, the following experiments were conducted based on MBDA-3 h. As can be seen in Figure 1b, which presents the ^1^H NMR spectrum of copolymer in CDCl_3_, the peaks corresponding to the protons of the double bonds completely disappearred. The assignment peaks are listed in Table 2. The peak at δ = 3.61 ppm was attributed to the methoxy protons (–OCH_3_) of PMMA (poly (methyl methacrylate)) and δ = 4.00 ppm was the resonance peak of the protons of the ethoxy (–OCH_2_–) on PBA (poly (butyl acrylate)). At δ = 6.99 was the proton resonance signal on amide N (–NH–) in PDAAM (poly (diacetone acrylamide)). However, the corresponding proton peak of AA carboxylate group at δ = 11.0 ppm was absent, which may be either due to the relatively low concentration of AA involved in copolymerization or plausibly arose from the inconvenient NMR resonance of acid monomer. The peaks at δ = 0.901 ppm were assigned to the protons of the methyl groups in the polymer that were not attached to the oxygen. Fourier transform-infrared (FT-IR) spectroscopy was used as the supplementary tool to confirm the copolymerization. As sketched in Figure 1c, the FT-IR analysis of MBDA copolymer was performed in the range from 4000 to 500 cm^−1^, and the illustration (Figure 1d) showed the confined wavenumbers from 1800 to 1550 cm^−1^ for pure monomers and copolymer in order to investigate the reaction between “C=C” characteristic bonds. The absorption band of "C=C" in 1650–1600 cm^−1^ was present in spectrums of all unreacted pure monomers except for the copolymer, which was another supporting evidence of successful copolymerization of monomers by forming linkages on the "C=C" backbone. Furthermore, other characteristic peaks were also found in the spectrum. For example, the region from 3558 to 3065 cm^−1^ was the overlapped characteristic peak of –NH– in DAAM and –OH in AA; the peak at 2931 cm^−1^ was the stretching vibration band of –CH_3_; the stronger absorption band at 1732 cm^−1^ was attributed to –C=O; the bending vibration band of –NH– with comparatively low intensity was still visible at 1609 cm^−1^; C–C–O and C–O–C showed characteristic absorption peaks at 1238 and 1142 cm^−1^, respectively. With reference to the interpreted data, it can be inferred that MMA, BA, DAAM, and AA had participated in the free radical copolymerization and the successful synthesis of MBDA copolymer was confirmed.

### 3.2. Characterizations of MBDA–TPP–NPs

The composite micelles, loading the water-insoluble porphyrin (TPP), dispersed stably in water by the hydrophilic group of MBDA copolymer, and the appearance of emulsion remained translucent blue despite TPP dosage up to 5 mg. The particle size and its distribution were determined by DLS, the concentration of the micellar emulsion was 0.2 mg/mL. As indicated in Figure 2a, the average particle sizes of micelles with different TPP contents were all below 25 nm, which are similar to previous studies of photothermal nanodots based on self-assembly of peptide−porphyrin conjugates [27]. In addition, it can be seen that the sizes of composite micelles were larger than that of pure micelles due to the padding of TPP. However, the mean particle size of micelles mildly decreased from 21.13 to 16.80 nm with the increase in TPP. In particular, this reasonable trend could be explained by the existence of a variety of interactions between TPP molecules and the micelle core. When they dispersed well in the emulsion but without agglomerates, the intensive interactions “bound” the loose chains and reduced the gaps inside micelle units. Therefore, the particle size decreased. The better dispersion of particles also corroborated this explanation. Atomic force microscopy (AFM) images have further confirmed that the particles exhibit the spherical shape and uniform size distribution (Figure 2b).

Figure 3a,b shows the distinct absorbing efficiency of different dispersions from 350 to 850 nm in wavenumber. For the pure emulsion without TPP and TPP in water, no characteristic absorption occurred. However, the micellar emulsion that included TPP expressed an obvious absorption peak at 417 nm (Soret band) and four peaks occurred at around 513, 546, 591, and 650 nm (Q band), respectively. This was consistent with the characteristic peaks of TPP in THF [28,29]. The conclusion drawn from the comparison of different dispersions is that the TPP-loaded micelles with red-light absorption were successfully prepared and had excellent dispersion in emulsion. The absorbance of TPP at 417 nm was tested by UV–Vis and employed to set up a standard function of TPP concentration vs. absorbance, which was summarized ([*y* = 6.9755*x* + 0.0244, *R*^2^ = 0.99992], *y* and *x* represent absorbance and TPP concentration (mg/100 mL), respectively). MBDA–TPP–NPs micelle solutions of 200 μL, with different TPP contents of 200 mg/mL, were dried completely and dissolved in 12 mL THF to prepare a solution with a concentration of 3.33 mg/mL. The absorbance of the solution at 417 nm was measured by UV–Vis absorption, and calculated by standard function. Accordingly, the concentrations of TPP in different micelles were 0.092 mg/100 mL, 0.277 mg/100 mL, and 0.376 mg/100 mL, and the corresponding TPP load factors were 92.98%, 93.69%, and 38.08%, respectively. All results are depicted in Figure 3c. The result showed a relatively high TPP loading efficiency belonging to MBDA-TPP 1.5-NPs. The photothermal effect of micelles is demonstrated in Figure 3d. The diagram shows the temperature changes of MBDA emulsion with exposure time irradiated by 655 nm laser (528 mW/cm^2^). The laser source was 5.5 cm away from the 500 μL micellar emulsion, the temperature is detected by an infrared thermometer. The result was that the increase in TPP loading and micellar concentration applied a beneficial effect on the rate of temperature rise and the reached temperature.

### 3.3. Characterizations of MBDA–TPP–ADH films

Glass transition temperature (*T*_g_) is one of the main features of SMP films because of its close connection with actuation temperature. In this article, differential scanning calorimetry (DSC) was performed to investigate the *T*_g_ of the SMP films. The *T*_g_ value was obtained from the second heating cycle measuring with a heating rate of 10 °C/min over the range from 20 to 85 °C under a nitrogen atmosphere. The MBDA–TPP–ADH films were prepared by cross-linking of MBDA–TPP–NPs loaded with TPP at different concentrations. The full DSC curves of various films were plotted in Figure 4a, in which *T*_g_ was around 46.8 °C for the film without TPP and 44–46 °C for the film with loading different dose of TPP. The DSC tests not only revealed the actuation temperature, but also indicated the condition for film deformation. A temporary-shaped, crack-free film was formed under an external force above the *T*_g_, as the phase was transforming from a glassy state to a rubbery state. Additionally, the UV–Vis absorption spectra of the SMP films (thickness: 0.3 mm) were sketched in Figure 4b. In contrast to non-absorbance in the film without TPP, the absorption peaks of different films loaded TPP in Soret and Q bands were highly consistent and displayed almost no shift as compared to TPP in THF [28], which represent the excellent dispersion of TPP in all films. Additionally, the absorbance efficiency tended to improve as the TPP dosage increased. More importantly, an optical transition at the wavelength of ~650 nm was demonstrated, providing data support for selecting the red light of the 655 nm laser as the radiation source for stimulating shape memory. In addition, more effective light trapping of the film resulted from the increased absorbance that, inevitably, strengthened the photothermal conversion, which facilitates the shape recovery of materials.

To further explore the thermal response of various films upon red light, we also investigated the effects of TPP doses and light intensity on the increased temperature of films, with results presented in Figure 5. The film fixed at a distance of 5.5 cm from the laser light source was directly irradiated with a beam of lamp (655 nm, 354 mW/cm^2^), and the temperature of the direct spot was continuously monitored by a Gasdna-ir 22 fixed infrared thermometer. Samples all started at room temperature. As shown in Figure 5a, distinguished from no change in temperature of the pure film without TPP, significant and rapid heating was observed in TPP loaded samples, which were characterized by original rapid increase followed by a gradual boost later of temperature. What is more, the increased temperature was dependent on both the dose of TPP and the light intensity. At the same light intensity, the temperature of the corresponding film boosted faster as the TPP increased. Figure 5b describes the phenomenon that the temperature rise rate was improved and the desired temperature was reached within 30 s when the light intensity increased, with the TPP fixed at 1.5 mg. These experiments further demonstrate the notion that the addition of TPP causes and improves the light capturing ability and the photothermal effect of the SMP film. Furthermore, the optimal result of the photothermal effect could be achieved by adjusting suitable light intensity and dosage of TPP.

### 3.4. Light-Actuated Shape Memory Effect

As shown in Figure 6a–c, the shape-memory behavior was first photo-recorded by exposing the bent film (under temporary shape) to a solid-state laser at 655 nm with a power of 214 mW/cm^2^. The shape recovery fulfilled by irradiating the bent position of the film against the laser beam. In contrast to the film unloaded TPP, the TPP-containing film exhibited a complete shape recovery process. An unfolded angle was the instantaneous ‘‘angle’’ in recovery determined by measuring the angle between the straight ends of the bent film. For more in-depth research on the light-active shape recovery progress, changes in the contraction force of films loaded with different amount of TPP under an extension of 140% (stretched length/original length × 100% = 140%) upon laser irradiation were recorded in Figure 6d. When exposed to the laser light, essentially no changes occurred in shape and stress for films without TPP. Significant and rapid shape recovery was observed in films with various TPP concentrations. The results demonstrated that TPP was essential to shape recovery because the photothermal effect of TPP increased the temperature of the film above its *T*_g_. Without TPP, any generated energy from laser exposure was insufficient to support the shape recovery, which in principle revealed a similar manner to the photothermal triggered shape memory polymer loaded with AuNPs [30].

Figure 7 demonstrates that the MBDA–TPP–ADH shape memory film prepared in this study possesses optically controllable spatially selective shape memory recovery. An external force was applied to the flat film at 80 °C, and the temporary shape of an octahedron was fixed by removing external force at room temperature (Appendix A). The octahedron was placed on a transparent cuvette with the edge of the octahedron exposed to direct light (655 nm, 528 mW/m^2^) to expand the specified side, opposite to the unexposed edge, which also indicated a time-controlled light-excited shape recovery. A plurality of intermediate states of the shape memory film between the temporary and initial shapes were achieved by controlling the switching of laser. When the laser was turned on, the local temperature of the exposed region rose above *T*_g_, and the restoring force generated by energy storage could relax the molecular chain and prompted the film recovery to original shape. The temperature was decreased below *T*_g_ after turning off the laser, and the freeze of molecular chains resulted in an intermediate state. The original planar shape received by exposing the whole film completely. The entire recovery process was recorded in Figure 7a. Similarly, Figure 7b reveals the same mechanism, but the film was exposed to light transmitted through the aqueous medium (Appendix A). The result showed that the aqueous medium applied no effect on shape recovery behavior and an unchanging temperature of water was detected. This exploration demonstrates the material obtained from the study is expected to be used in aqueous media.

To state the ability of this material for use “within an aqueous solution”, the film (7.45 × 6.80 × 0.30 mm) was submerged in water prior to light activation, three times. Figure 7c shows the unfolded angle of MBDA–TPP 1.5–ADH and MBDA–TPP 0–ADH films, immersed with exposure time within 40 s (528 mW/m^2^). Compared with film without TPP, MBDA–TPP 1.5–ADH film exhibited a superior shape recovery upon the light irradiation, the recovery speed of the first 10 s was fast, and then tended to be gentle. After cumulative exposure time of 40 s, the recovery rate of the film reached ~70%. In addition, it is worth to mention that no significant fluctuations (33.3–33.8 °C) were found for the temperature of water environment, indicating that the thermal tissue damage caused by the heat released from the material was negligible.

## 4. Conclusions

A porphyrin-loaded copolymer material has been prepared by copolymerization of monomeric MMA, AA, BA, and DAAM through self-assembly, whose shape memory effect could be driven in a non-contact manner by red light. Based on the photothermal effect caused by TPP under the optical radiation, the temperature of the region rose above *T*_g_, and the relaxation (or strain energy release) of the molecular chain caused the film to recover to original shape. Additionally, the recovery rate and degree were closely related to TPP loading. Moreover, the achieved material possessed both spatial and temporal control in shape memory effect. An optimal means for heating (e.g., transdermal light exposure to heat implants) provided by red light helps apply this material in a wide range of fields.

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
