# Peer review of "Photothermal-Triggered Shape Memory Polymer Prepared by Cross-Linking Porphyrin-Loaded Micellar Particles"

_materials, 2019, doi:10.3390/ma12030496_

Round 1

Reviewer 1 Report

In their report "Photothermal-triggered shape memory polymer prepared by cross-linking porphyrin-loaded micellar particles" Qian et al. have developed a novel shape memory polymer film possibly useful for biomedical application. This development is interesting, the characterization is extensive and appropriate and the paper is overall well written. However, I suggest the following changes to increase the quality of the paper.

1. At several points the authors suggest biomedical application due to biocompatibility of porphyrin. This could become more concrete: what exactly are the applications the polymer could/should be used?

2. Scheme 1 is very detailed at very low resolution making it hardly readable/visible.

3. The Results/Discussion part does not include much of a discussion. It does neither reference back to literature from the Introduction part nor does it include any additional reference. This should be improved by discussing the findings with respect to the existing literature.

Author Response

Dear,

    The response is uploaded as a word file.

Regards,

Xiaorong,Wang

Reviewer 2 Report

The manuscript should be reviewed for proper English grammar and punctuation.

Figure and Scheme labels and text are too small, and overall the layouts are difficult to read

Mn and Mw should use subscripts, Tg should use an italicized subscript

Section 3.4: Discussion around the actuation method is not clear. Please re-write into a more readable format.  The discussion of the angles is not clear and should be re-written for reader clarity. Overall, this section is very difficult to comprehend, in part due to the lack of punctuation as well as the poor explanation of concepts such as 140% strain. Additionally, the use of phrases such as dramatic increase are difficult to accept, as no errors are included in the plots. Finally, the manuscript specifically refers to the plots themselves, rather than just the data, and the removal of discussion of graphics and plots will both reduce manuscript length and improve section clarity.

Figure 6 images are mostly background, and as such as not overly useful for the reader. Consider increasing the size of the polymer relative to the background space.

Figure 7 and the corresponding text claim spatial control of shape recovery, which is not demonstrated, or at least is not demonstrated well enough to be found examining the text and figure. It would seem that the chosen demonstrations (both the bent film and the octahedron) could be achieved through simple heating as well as through single exposure of the film (does the film act as a waveguide here, interacting with the absorption?). Little evidence is provided that porphyrin particles are the source of the recovery in this manner, and subsequent studies would be useful in providing justification for the authors' claims.

Based upon the provided experiments, it is suggested that subsequent testing focus on presenting limited exposure of light to a film with multiple mobile sections, to determine light transmittance (spatial control) and possible diffusion of light throughout the film. Additionally, data on temperature changes during the exposure would be useful.

Conclusions: While the reviewer is not entirely certain of this, it is very likely that the phrase "we have invented" is incorrect (legally and academically) and should therefore be changed; the use of composite polymers is not novel, nor are photo-responsive composites. Use a different phrase. Biocompatibility and degradability have in no way been shown and should therefore be removed from the conclusions. Overall, this section makes claims that aren't supported by the data provided. While a number of applications could make use of this technology, the experiments do not support the claims made by the authors. Revise this to be more inline with what has been provided.

Author Response

Dear,

  The response is uploaded as a word file.

Regards,

Xiaorong, Wang

Reviewer 3 Report

The authors present a novel co-polymer embedded with a photoactive organic compound (Porphyrin-loaded MBDA micelle).  The work is unique in its utilization of natural materials to photothermally active the polymer construct.  Methods used in characterization of the synthesis and material properties were well presented.  The results proved a promising shape memory material under the given tested environments (submerged in THF and dried in air).

On that note, the claim for utility as a biocompatible component was not fully explored.  To make this definitive assertion, a cytotoxicity test (ISO 10993-5) would need to be performed on the material to confirm non-toxic levels of residual THF.  

Further, Figure 7 shows that the material actuation can occur following light application ‘through an aqueous medium’.  However, to state the ability of the material for use ‘within an aqueous solution’, the experiment should be repeated with the material submerged prior to light activation. 

A side note to the authors for future experiments may include the level of heat released from the material during shape recovery while activated in situ (aqueous solution) to evaluate its propensity for thermal tissue damage as well.

Overall, the work presented is well constructed and presented overall.  The topic is contemporary in its use of natural materials and interesting for biomaterials research.

Author Response

(The authors gave the same response as above.)
